# Questionnaire-Based Survey during COVID-19 Vaccination on the Prevalence of Elderly’s Migraine, Chronic Daily Headache, and Medication-Overuse Headache in One Japanese City—Itoigawa Hisui Study

**DOI:** 10.3390/jcm11164707

**Published:** 2022-08-11

**Authors:** Masahito Katsuki, Junko Kawahara, Yasuhiko Matsumori, Chinami Yamagishi, Akihito Koh, Shin Kawamura, Kenta Kashiwagi, Tomohiro Kito, Akio Entani, Toshiko Yamamoto, Miyako Otake, Takashi Ikeda, Fuminori Yamagishi

**Affiliations:** 1Department of Neurosurgery, Itoigawa General Hospital, Itoigawa 941-0006, Niigata, Japan; 2Department of Health Promotion, Itoigawa City, Itoigawa 941-8501, Niigata, Japan; 3Sendai Headache and Neurology Clinic, Sendai 982-0014, Miyagi, Japan; 4Department of Neurology, Itoigawa General Hospital, Itoigawa 941-0006, Niigata, Japan; 5Department of Neurosurgery, Nou National Health Insurance Clinic, Itoigawa 949-1331, Niigata, Japan; 6Department of Internal Medicine, Itoigawa General Hospital, Itoigawa 941-0006, Niigata, Japan; 7Department of Nursing, Itoigawa General Hospital, Itoigawa 941-0006, Niigata, Japan; 8Department of Surgery, Itoigawa General Hospital, Itoigawa 941-0006, Niigata, Japan

**Keywords:** aged, artificial intelligence, chronic daily headache (CDH), cluster analysis, epidemiology, medication-overuse headache (MOH), migraine, prevalence

## Abstract

Background: The prevalence of headache disorders, migraine, chronic daily headache (CDH), and medication-overuse headache (MOH) among the elderly in Japan has not been sufficiently investigated. We performed a questionnaire-based survey and revealed 3-month headache prevalence and headaches’ characteristics. Methods: The population aged over 64 was investigated in Itoigawa during their third coronavirus disease 2019 vaccination. Migraine, MOH was defined as The International Classification of Headache Disorders Third edition. CDH was defined as a headache occurring at least 15 days per month. K-means++ were used to perform clustering. Results: Among 2858 valid responses, headache disorders, migraine, CDH, and MOH prevalence was 11.97%, 0.91%, 1.57%, and 0.70%, respectively. Combined-analgesic and non-opioid analgesic were widely used. Only one migraineur used prophylactic medication. We performed k-means++ to group the 332 MOH patients into four clusters. Cluster 1 seemed to have tension-type headache-like headache characteristics, cluster 2 seemed to have MOH-like headache characteristics, cluster 3 seemed to have severe headaches with comorbidities such as dyslipidemia, stroke, and depression, and cluster 4 seemed to have migraine-like headache characteristics with photophobia and phonophobia. Conclusions: This is the largest prevalence survey in the Japanese elderly. Headache disorders are still the elderly’s burden. Clustering suggested that severe headaches associated with some comorbidities may be unique to the elderly.

## 1. Introduction

Headache is a common public health problem [1,2]. The two representative primary headaches are migraine and tension-type headache (TTH), and they are described in the International Classification of Headache Disorders 3rd edition (ICHD-3) [3]. In Japan, the total prevalence of migraine is 4.3–8.4%, and 74.2% complain that migraine attacks impair their lives [4,5,6,7]. Japanese migraineurs have an incremental burden compared to non-migraineurs in terms of decreased health-related quality of life, impaired productivity, more healthcare provider visits, and higher indirect costs [8]. Additionally, about 15–20% of Japanese individuals have TTH, and 22.4–29.2% complained that TTH disturbed their performances [9,10]. However, 59.4% of patients with primary headaches had never consulted a physician about their headache disorders [9]. Therefore, the majority of headache sufferers presumably manage the pains by taking over-the-counter (OTC) medicines. In addition, only neuroimaging is performed when headache sufferers visit a doctor to exclude emergent or secondary headaches, and the appropriate diagnosis of primary headache subtypes and their treatment is insufficient. Even when the doctors diagnose primary headaches, their treatment knowledge is unsatisfactory, leading to patient dissatisfaction [11]. These inadequate headache medical resources and OTC medicine use may lead to medication-overuse headache (MOH) and chronic migraine development [12].

Usually, the incidence and prevalence of migraine tend to decrease with age. However, these insufficient headache medical resources can cause MOH and chronic migraine in the elderly. Additionally, other diseases causing headaches, such as transient ischemic attacks and amyloid angiopathy, and the presence of multiple comorbidities and polypharmacy make the elderly’s headache disorders complicated [13]. 

Large-scale headache disorders prevalence studies were conducted in Japan in 1997 [7] and 2004 [6]. However, the social environment has changed over time, and there have been few prevalence studies of headache disorders in Japan in recent years. We performed a MOH prevalence survey among the working-age population (15–64 years old) during the first COVID-19 vaccination in 2021 [4], revealing that migraine prevalence was 4.26% and that of MOH was 2.32%. However, we only investigated the working-age population. Therefore, headache [14,15], migraine [15,16,17], chronic daily headache (CDH) [14,15,17,18,19,20,21,22], and MOH [12,14,15,17,19,21] prevalence among the elderly in recent years was not sufficiently investigated by a population-based survey in Japan. CDH in this context means as a case with headaches occurring at least 15 days per month for three or more consecutive months [14,15,17,18,19,20,21,22].

Furthermore, the ICHD-3 is a clinically useful diagnostic criterion for headache disorders, but it may not always clearly separate the headache disorders characteristics, and multiple diagnoses are sometimes applied. Moreover, ICHD-3 cannot always diagnose all headache disorders [23]. For example, not all of the CDH defined by Silberstein [24] can be classified according to ICHD-3. The definition and criteria, especially chronic migraine and transformed migraine (TM), have been discussed [25]. From this experience, we investigated CDH prevalence and hypothesized that there may be further headache disorders subgroups specific to the elderly, which cannot be grouped by ICHD-3.

In this context, we performed a questionnaire-based survey on the elderly’s headache disorders prevalence. The primary objectives of this project were to clarify the headache, migraine, CDH, and MOH prevalence in the Japanese elderly aged over 64 years. The secondary objective was to identify the characteristics of the elderly’s headache disorders. The third objective was to identify the mathematically hypothesized headache disorders’ subgroups in the elderly using clustering methods of k-means++ [26].

## 2. Materials and Methods

### 2.1. Study Population and Survey Procedure

We performed this cross-sectional questionnaire-based study during the 15-min waiting period after the third COVID-19 vaccination, with sufficient infection control like sterilization. The Japanese vaccination law was revised on 1 December 2021, and all the Japanese people have received the third COVID-19 vaccination as a duty of effort. It is not mandatory, so people had the right to refuse. However, the third COVID-19 vaccination coverage among the elderly was 90.1% on 20 July (https://www.kantei.go.jp/jp/headline/kansensho/vaccine.html, accessed on 21 July 2022). In Itoigawa city, the first vaccination started in June 2021, and the second vaccinations were completed in November 2021. There were two large vaccination sites (Itoigawa General Hospital and Nou National Health Insurance Clinic) and 11 other small vaccination sites, such as clinics. We performed this questionnaire at the two large vaccination sites during the third-time vaccination. The third vaccination started in March 2022 and was completed in June 2022. The citizens could select where they undergo vaccination. 

We handed the questionnaire sheet and a pen to the citizens who underwent the third vaccination, and they read it, wrote down the answers or filled in all the items on the questionnaire sheet. The items we asked are shown in Table 1 in the next paragraph. 

The inclusion criteria were: people aged over 64 with valid responses that filled in all the items on the questionnaire sheet. The exclusion criteria were: people who could not understand the questionnaire due to dementia, psychiatric disorder, or mental retardation, people who indicated that they did not want to participate in this study, and people with invalid responses as the questionnaire sheets with one or more blank answers. The diagnosis of dementia, psychiatric disorders, and mental retardation was self-reported by respondents or their families. People aged over 64 who met these criteria were analyzed. Among those, headache, migraine, CDH, MOH prevalence, and their relationship to the items on the questionnaire sheet were investigated.

### 2.2. Items in the Questionnaire Sheet

The questionnaire sheet consisted of the following 13 items: age, sex, comorbidities, modified Rankin Scale (mRS) [27], how many days per month headache occurs in these 3 months or no headaches, characteristics as (1) unilateral location, (2) pulsating quality, (3) moderate or severe pain intensity, (4) aggravation by or causing avoidance of routine physical activity, (5) nausea and/or vomiting, (6) photophobia and phonophobia, the headache duration, what acute medication you use, how many days per month you use the acute medication, use of prophylactic medication for headache, and what prophylactic medication you use (Table 1).

### 2.3. Definition of Migraine, CDH, and MOH

In the valid respondents, cases with headaches within the last 3 months were considered headache sufferers. Migraine and MOH were defined in compliance with the ICHD-3 [3] as much as possible in the questionnaire sheet, but not strictly the same as described in the ICHD-3.

A case of migraine in this study was defined as a respondent, with chronic headache which lasts 4–72 h, who had at least 2 of the characteristics; (1) unilateral location, (2) pulsating quality, (3) moderate or severe pain intensity, (4) aggravation by or causing avoidance of routine physical activity and had nausea and/or vomiting OR photophobia and phonophobia. 

A case of MOH was defined as a respondent who had headaches ≥15 days per month and reported intake of acetaminophen ≥15 days per month, combination analgesic (most OTC medicines) or triptan ≥10 days per month. These case definitions approximate Criteria B, C, and D of the ICHD-3 diagnostic criteria for migraine without aura (code 1.1) and Criteria A and B of the ICHD-3 diagnostic criteria for MOH (code 8.2). Also, CDH was defined as a case with headaches occurring at least 15 days per month for 3 or more consecutive months [28].

### 2.4. Statistical Analysis and Clustering by Artificial Intelligence

The Shapiro–Wilk test was used to check the normal distribution, and the results are shown as median (interquartile range; IQR) for the variables with non-normal distributions. The population proportion test evaluated the prevalence. We used chi-square, Fisher’s exact, or Mann–Whitney U test was performed to compare the 2 groups. The Kruskal–Wallis test was used to compare the characteristics among the multiple clusters described later. A two-tailed *p* < 0.050 was defined as statistically significant without Bonferroni correction [29].

We performed a non-hierarchical clustering method to classify the headache disorders of the elderly with mRS 0–2. We used 25 vectors obtained as all the answers to the questionnaire sheet to perform k-means++ clustering [26]. The number of clusters was decided using the elbow chart and silhouette score. The differences in the variables among each cluster were investigated. We used SPSS software version 28.0.0. (IBM, New York, NY, USA) for statistical analysis and stratified clustering. Python 3.9.0 (Wilmington, DE, USA), scikit-learn 0.24.1, and Matplotlib 3.4.3 for k-means++ and calculating silhouette score [4].

### 2.5. Ethical Aspects

The study was approved by Itoigawa General Hospital Ethics Committee (approval number 2021-15). The questionnaire was anonymous and did not contain any personally identifiable information. The purpose of the study was explained to the subjects in writing and handed to them. If they could participate in the study, they were asked to complete the headache questionnaire. If they were unable or did not want to participate, they were asked to submit a blank sheet of questionnaire or not to receive the questionnaire sheet, thus providing an opportunity for non-participation. All methods were carried out under relevant guidelines and regulations (Declaration of Helsinki). This study deleted all personal patient information from the database to protect patient privacy.

## 3. Results

### 3.1. Prevalence of Headache Disorders

From the 5258 citizens aged over 64 years old who underwent the third-time vaccination in the 2 large vaccination sites, we acquired 2858 (54.35%) valid responses, including 1415 men and 1443 women. Of the 2858 respondents, 2620 (91.67%) individuals lived without help with mRS 0–2. At the end of March 2022, the overall population of Itoigawa City is 40,179, and the population aged over 64 is 16,378 (40.76%). Of the 16,378 elderlies, 13,412 (81.89%) live without help, which was revealed by a door-to-door survey by Itoigawa Health Care Center (Figure 1). This means that we collected the responses from 17.45% of all the elderly aged over 64 years and 19.53% of the individuals aged over 64 years old as well as who can live without help in Itoigawa City. 

Among the 2858 valid respondents with all mRS scores, 342 (11.97%) reported having experienced headaches in the last 3 months. Of them, migraine was reported by 26 (0.91%) respondents, CDH by 45 (1.57%), and MOH by 20 (0.70%). The prevalence of chronic headache and migraine peaked in the group aged 65–69 y.o. and 90 ≤ y.o., whereas that of MOH in the group aged 70–74 y.o. (Figure 1). Women tended to have a chronic headache (80.11%), migraine (92.30%), and MOH (90.00%) (Table 2). Of the 45 CDH respondents, nobody had migraine, but 9 (20.00%) had MOH.

Among the 2620 valid respondents with mRS 0–2, 332 (12.73%) reported having experienced headaches in the last 3 months. Two-hundred sixty-six of them (80.12%) were women, and the median age (IQR) was 70 (66–76) y.o. Of them, migraine was reported by 24 (0.92%) respondents, CDH by 44 (1.69%), and MOH by 19 (0.72%). (Table 2). Of the 45 CDH respondents, nobody had migraine, but 9 (20.45%) had MOH.

### 3.2. Characteristics of Elderly’s Migraine and MOH

Of the mRS 0–2 respondents, 24 (7.23%) of the all-headache disorders respondents had migraine. Twenty-two were women (91.67%), and their median age was 70 (62–72). The median headache-occurring days was eight (2–16) per month. All the migraineurs’ characteristics had unilateral location. Eighteen (75.00%) respondents’ migraines last 9–24 h. The median acute medication use days per month was one (0–2) (Table 3). Of the acute medication, 11 (45.80%) used combination-analgesic, 2 (8.30%) used loxoprofen, 1 (4.20%) used kampo, as answers to multiple-choice questions. However, 10 (41.70%) did not use acute medication. Only one (4.17%) individual used prophylactic medication of Japanese herbal kampo goreisan [18,30,31] (Figure 2). Ten (41.70%) had hypertension, two (8.3%) had dyslipidemia, and the other two had back or knee pains (Figure 3).

Of the mRS 0–2 respondents, 19 (5.72%) of the all-headache disorder respondents had MOH. Seventeen were women (89.47%), and their median age was 68 (65–70). The median headache-occurring days was 28 (16–30) per month. Three (15.70%) had migraine. Ten (52.63%) respondents’ headaches last 1–3 h, and others’ last 9–14 h. The median acute medication use days per month was 20 (10–30) (Table 3). Of the acute medication, 14 (73.70%) used combination-analgesic, 4 (21.00%) used loxoprofen, and 1 (5.30%) used acetaminophen, as answers to multiple-choice questions. Nobody used prophylactic medications (Figure 2). Four (21.10%) had dyslipidemia, another four had back or knee pains, three (15.80%) had hypertension, and two (10.50%) had heart disease or gastrointestinal disease (Figure 3). Of the 19 MOH patients, the number of estimated MOH diagnoses were as follows; 3 (15.79%) non-opioid analgesic-overuse headache (ICHD-3 code 8.2.3), 16 (84.21%) combination-analgesic-overuse headache (code 8.2.5).

### 3.3. Clustering Results

We treated the all-headache disorders respondents with mRS 0–2. The 25 variables (age, sex, the presence of 13 comorbidities, headache frequency, six headache disorders characteristics, duration, frequency of acute medication oral intake, and use of prophylactic medication) were used for the k-means++ clustering. The elbow chart suggested that four to six is the appropriate number of clusters (Figure 4A). The silhouette score was calculated, and that of four clusters was highest as 0.5704 (Figure 4B). Therefore, we decided the number of clusters as four. The mean values of each cluster are shown in Table 4, and the clusters are plotted in the three-dimensional space. The axes were calculated by the principal component analysis consisting of the 25 variables (Figure 4C and Appendix A).

Finally, we performed the Kruskal–Wallis test and chi-square test to compare the characteristics of the 4 clusters. Sex, presence of dyslipidemia, stroke, depression, headache frequency, headache disorders characteristics of unilateral location, pulsating quality, photophobia and phonophobia, aggravation by or causing avoidance of routine physical activity, moderate or severe pain intensity, headache duration, and frequency of acute medication oral intake were significantly different (Table 4). From the results, cluster 1 seemed to have TTH-like headache characteristics, cluster 2 seemed to have MOH-like headache characteristics, cluster 3 seemed to have severe headaches with comorbidities such as stroke and depression, and cluster 4 seemed to have migraine-like headache characteristics with photophobia and phonophobia.

## 4. Discussion

We report that the headache disorders, migraine, CDH, and MOH prevalence among the elderly aged over 64 years in Itoigawa was 11.97%, 0.91%, 1.57%, and 0.70%, respectively. We hypothesized that the elderly’s headache disorders could be grouped into four clusters; (1) TTH-like headache characteristics, (2) MOH-like headache characteristics, (3) severe headaches with comorbidities such as stroke and depression, and (4) migraine-like headache characteristics with photophobia and phonophobia.

### 4.1. Headache Disorders Prevalence in Elderly

Previously, 11 population-based surveys [6,7,12,14,15,16,17,19,20,21,22], 2 headache clinic surveys [32,33], and 2 insurance data surveys [34,35] were conducted to investigate the headache disorder prevalence in the elderly (Table 5). The headache disorders prevalence was about 12–76%, migraine prevalence 1–11%, CDH prevalence 1.6–53.4%, and MOH prevalence 0.70–2.60% worldwide. Our relatively small prevalence of headache disorders may be because the 3-month prevalence was examined in this study considering ICHD-3 criteria rather than the 1-year prevalence. Additionally, the results depending on the country and study design and recall bias should be considered. However, our study is the third report on the elderly’s headache disorders prevalence and the first on MOH prevalence in Japan, with a large population.

Our previous study on the headache disorder prevalence in the same region among the working age revealed that the headache disorders, migraine, and MOH prevalence is 41.05%, 4.26%, and 2.32% [4]. On the other hand, those among the elderly were 11.97%, 0.91%, and 0.70%. Considering these results, this study on the elderly’s prevalence suggested similar results: the prevalence of headache disorders progressively decreases with advancing age, which is still consistent more than 20 years later [6,7], although healthy life expectancy becomes longer.

### 4.2. Migraine in Elderly

Migraine generally begins during early childhood with a peak around puberty. A small number of people first self-report in their sixth, seventh, and eighth decades of life. Indeed, migraine is most active between the third and fourth decades of life with the majority of elderly chronic migraine patients indicating an onset of migraine prior to 50 years. Importantly, given the worldwide increase in life expectancy, older age migraine is likely to become a far greater personal and public health issue over the next 40 years as management is likely to be confounded by other health problems and consequent association with polypharmacy [13,36].

In Japan, where headache disorder treatment is still inadequate, treatment for elderly’s migraine is presumed to be even more inadequate [5,11]. The clinical characteristics of migraine change with age and as comorbidities increase. The percentage of men with migraine decreases markedly with age. In women, duration of headache, unilateral pain, pulsating sensation, light sensitivity, and noise sensitivity increases with age. In contrast, worsening of headaches with physical activity decreased with age. Neck pain also increases with acute attacks of migraine in the elderly [13]. These changes of migraine symptoms and the stigma that migraine is rare in the elderly and that any headaches with neck pain is often considered as tension-type headache [37] may lead to misdiagnosis by doctors, which can be a major burden for migraine patients, who account for about 1% of the elderly population. In addition, nausea, vomiting, and difficulty with physical activity in the elderly can lead to dehydration and disuse syndrome. Therefore, our results will inform the doctors of the importance of treatment for and burden of elderly migraine in an aging society with a declining birthrate.

### 4.3. CDH in Elderly

We showed that there were 1.57% of CDH elderlies with mRS 0–4 and 1.69% with mRS 0–2, while those with migraine (0.91% for mRS 0–4 and 0.92% for mRS 0–2) and MOH (0.70% for mRS 0–4 and 0.72% for mRS 0–2) had low prevalence compared to CDH. CDH represents a range of disorders characterized by long-duration headaches 15 or more days per month. Silberstein proposed that CDH is a group of disorders including chronic TTH (CTTH), TM, new daily persistent headache, and hemicrania continua [38]. Of the four subgroups of CDH, TM is now regarded as chronic migraine or episodic migraine with MOH [11], but there is a discrepancy between TM and CM criteria [25]. TM is characterized by typical migraine headaches with recurrent attacks in the first 10 to 20 years of age. However, from middle age, the frequency of headaches increases to daily or almost daily. The severity of headaches decreases, and the pain becomes similar to TTH. Photophobia, phonophobia, and nausea become less prominent. However, the migraine component persists, such as exacerbation during menstrual periods and unilateral, pulsating characteristics [39]. Considering these changes of headache characteristics of TM from EM with a long time course, the elderly’s CDH might include TM misdiagnosed as CTTH. We should consider the probability of TM or CM when we see CDH elderly and should avoid a stupid diagnosis of CTTH without keeping a prospective headache diary. Further discussion on the diagnostic criteria of TM and CM is needed [25,40].

### 4.4. MOH in Elderly

Previously, 6 population-based surveys [12,14,15,17,19,21] on the elderly’s MOH prevalence were performed. The prevalence was reported as 0.98–2.60%. Our results of the elderly’s MOH prevalence of 0.70% seems a bit small compared to other countries’ reports. This could be due to the fact that lifestyle-related diseases are now being properly treated so that more people are taking antihypertensive drugs that also function as migraine prophylactics, and people are becoming more educated over time [41]. Another possibility is that Japanese herbal kampo medicines are commonly prescribed for the elderly in Japan and used as an acute alternative therapy for headache disorders [18,30,42], preventing the abuse of common analgesics. In addition, the fact that the elderly have more TTH [6,7] and often tolerate it until it improves without the use of acute medications may reduce the rate of MOH development.

On the other hand, low physical activity [21] and comorbidities such as other pain syndromes, sleep-related disorders, and gastrointestinal disorders, which aggravate migraine [43], supposedly lead to MOH development. Additionally, mild cognitive impairment can make it difficult to keep headache diaries and can make medication management impossible. Therefore, appropriate guidance and supportive care are needed not only for the elderly patients but also for their family members to prevent medication overuse. Further awareness of headache disorders, migraine, and MOH is needed for the elderly and their families.

### 4.5. Clustering Results

Clustering is the division of a set of classification objects into subsets so that internal cohesion and external isolation are achieved, providing insight into the differences in patients’ phenotypes, independent of existing diagnoses (unsupervised learning). There are four previous reports [4,44,45,46] using the clustering method, but there are no reports on clustering the elderly’s headache disorders characteristics. 

In our study, we performed the k-means++ to group the 332 elderly headache disorders into four clusters: (1) TTH-like headache characteristics, (2) MOH-like headache characteristics, (3) severe headaches with comorbidities such as stroke and depression, and (4) migraine-like headache characteristics with photophobia and phonophobia. Our result suggests two important issues: (1) The clusters were relatively clearly separated shown by Appendix A. On the other hand, it also suggested that headache disorders among the elderly was heterogeneous and had a spectrum. (2) Although Clusters 1, 2, and 4 seemed to be relatively consistent with the diagnostic criteria of ICHD-3, Cluster 3 was a group with many comorbidities and provided a new perspective on headache disorders in the elderly. Headache disorders are heterogeneous diseases, and our clustering results confirmed that the ICHD-3 can diagnose headache disorders to some extent, but it does not always classify it neatly.

Cluster 3 is a group with many comorbidities, such as dyslipidemia, stroke, and depression. A systematic review revealed the elevated risk of migraine in obese individuals compared to normal-weight persons [47]. The secretion of several inflammatory-related proteins from adipocytes may be related to headache disorders. On the other hand, statin, which is commonly used for dyslipidemia treatment, may have a prophylactic effect on migraine, supposedly via anti-inflammatory and vasodilatory effects [48]. Statins themselves may suppress migraine headaches, but statins are sometimes routinely prescribed for ischemic stroke patients [49]. Therefore, we focus here on the relationship between stroke and headache disorders. Persistent headache attributed to past ischemic stroke (ICHD-3 code 6.1.1.2) and headache attributed to transient ischemic attack (TIA) (code 6.1.2.) are described in ICHD-3. Cerebrovascular endothelial dysfunction may be related to a proinflammatory effect, causing migraine [43]. Additionally, cilostazol is prescribed for ischemic stroke to prevent stroke recurrence. It has a smaller risk of bleeding but a side effect of headache [50]. A patient may have persistent pain after craniotomy due to a cerebral hemorrhage or subarachnoid hemorrhage—persistent headache attributed to a craniotomy (code 5.6). Even if patients are independent in their post-stroke lives, they may still be under stress due to some disability associated with depression. Moreover, aging itself is another cause of depression. Depression is up to 2.5 times more prevalent in patients with migraines than in the general population, with 40% reporting depressive episodes during their lifetime [51]. Given the potential explanation of the comorbidity between migraine and depression, the existence of a bidirectional relationship, such as serotonin, serotonin transporter gene polymorphism, dopamine receptor genotype, and gamma-aminobutyric acid [52], is possible. Cluster 3 is a severe headache disorder associated with these comorbidities and may be unique to the elderly. Unlike migraine in the young, it may require comprehensive support and treatment. Further research on the elderly’s headache disorder characteristics and classification are needed.

### 4.6. Insufficient Medical Resources for Headache Treatment

Itoigawa city, where this study was performed, is rural, and the medical resources are not rich. In rural areas with limited medical resources, the potential misuse of pain and headache clinics can be raised, seeing them as “last resorts” for hopeless cases [53]. A lack of appropriate treatment for chronic pain also remains an ongoing major healthcare problem in other countries. In Germany, Italy, and Turkey, doctors treating chronic cancer and musculoskeletal pain feel the need to change treatment for a third of the patients. They think of changing the pain medication (52.4%), co-medication (42.2%), and non-medical therapy (19.0%) [54]. Therefore, improved communication between patients and doctors is still needed to find the best medications [55]. From the Italian survey, doctors treating musculoskeletal pain do not frequently use multidimensional questionnaires to evaluate patients’ status, which assess physical, social activities, psychological distress, sleep, enjoyment of life, and functioning. This insufficient medical attitude towards pain treatment among primary care providers suggests the need of further efforts toward a pain management-oriented education for doctors and developed medical coordination systems between general practitioners and second-level pain centers [56]. These problems are also seen in Japan [57]. Additionally, a survey in Japan showed that over half of Japanese people incorrectly believe that they should endure pain, and that over 80% of them are unaware of appropriate ways to cope with pain [58]. About 20% of headache sufferers feel that doctors would not be kind to them if they consulted them. About 41% of them do not know the presence of prophylactic medications. To solve these problems, a headache awareness campaign for patients, non-headache sufferers, and doctors has also been performed [59]. However, the insufficient medical resources in rural Itoigawa and doctors’ and patients’ inappropriate attitudes toward their headaches may have influenced our survey result; this is one of the limitations.

### 4.7. Limitation

First, this study was performed in a rural city in Japan, and the respondents covered about only 20% of the elderly population. Itoigawa City does not yet have adequate neurological and headache disorders care resources, and the prevalence may vary compared to urban areas with more medical resources in Japan. Second, we performed this questionnaire at the two large vaccination sites, so we could not evaluate the citizens who did not come to the vaccination site or who selected other vaccination sites. Third, the recall bias can be present, affecting the prevalence results. Fourth, migraine and MOH should be diagnosed after filling a prospective 3-month headache diary. Our study investigated retrospectively, so the questionnaire-based diagnosis should be interpreted carefully. Additionally, the headache diagnosis was not strictly based on the ICHD-3; we did not investigate whether there were at least five migraine attacks or not. In addition, MOH is a headache occurring on 15 or more days/month in a patient with a pre-existing primary headache and developing as a consequence of regular overuse of acute or symptomatic headache medication (on 10 or more or 15 or more days/month, depending on the medication) for more than 3 months. However, we did not investigate the duration of regular overuse. Fifth, the four clusters of the clustering results in this study were mathematically decided, so the clinical meanings of each cluster should be carefully interpreted. Sixth, we could not investigate the possibility of other secondary headaches by the questionnaire sheet, which must be excluded by definition to reach a diagnosis of primary headaches based on the ICHD-3. Additionally, the use of medicines with side effects of headaches was not investigated. Furthermore, COVID-19 vaccination-related headaches and post-COVID-19 headaches were not investigated. Seventh, we did not ask about the presence of chronic obstructive pulmonary disease and sleep apnea, which are known comorbidities related to headaches. A further large study on the elderly’s headache disorders prevalence all over Japan is needed.

## 5. Conclusions

Headache disorders, migraine, CDH, and MOH prevalence among the elderly aged over 64 years in Itoigawa was 11.97%, 0.91%, 1.57%, and 0.70%, respectively. Combined-analgesic (most OTC medicines) and non-opioid analgesic (loxoprofen or acetaminophen) were widely used. Nobody used prophylactic medications for migraine. We performed k-means++ to group the 332 MOH patients into four clusters. Cluster 1 seemed to have TTH-like headache characteristics, cluster 2 seemed to have MOH-like headache characteristics, cluster 3 seemed to have severe headaches with comorbidities such as stroke and depression, and cluster 4 seemed to have migraine-like headache characteristics with photophobia and phonophobia.

## Figures and Tables

**Figure 1 jcm-11-04707-f001:**
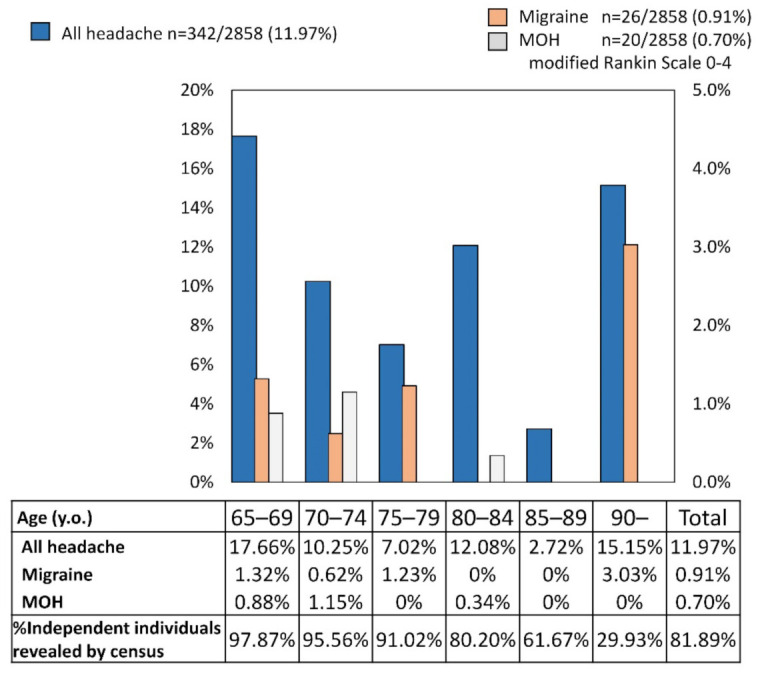
Age distribution of chronic headache, migraine, and medication overuse headache (MOH) in respondents aged over 64 with modified Rankin Scale 0–4.

**Figure 2 jcm-11-04707-f002:**
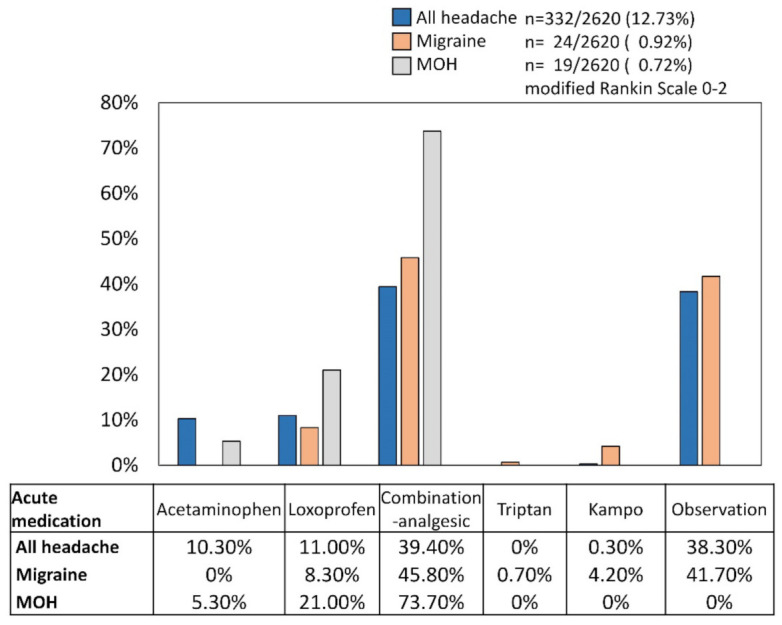
The proportion of acute medication use in all headache disorders, migraine, and medication overuse headache (MOH) respondents. The respondents all lived independently (modified Rankin Scale 0–2).

**Figure 3 jcm-11-04707-f003:**
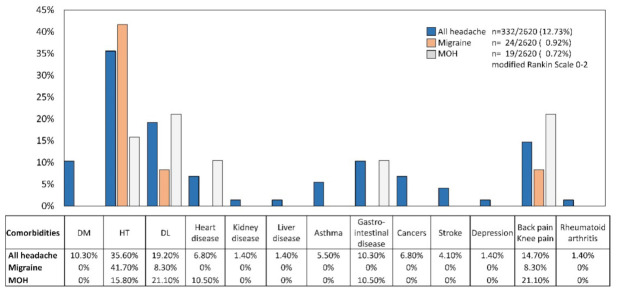
The proportion of comorbidities in all headache disorders, migraine, and medication overuse headache (MOH) respondents. The respondents all lived independently (modified Rankin Scale 0–2).

**Figure 4 jcm-11-04707-f004:**
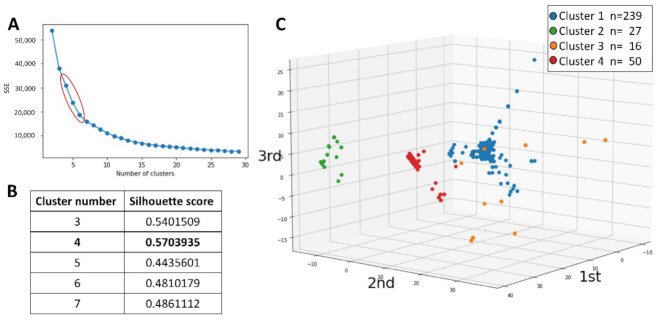
Results of clustering. (**A**): The elbow chart calculated by k-means++, suggesting four to six is the appropriate number of clusters. (**B**): The silhouette score was calculated, and that of the 4 clusters was highest as 0.5704. (**C**): The clusters are plotted in the three-dimensional space, of which the axes were calculated by the principal component analysis consisting of the 25 variables. SSE; sum of squared errors.

**Table 1 jcm-11-04707-t001:** Headache questionnaire sheet.

Questions	Answers
1. Age	( ) y.o.
2. Sex	Man or Woman
3. Do you have any diseases treated by doctors? diabetes, hypertension, dyslipidemia, heart disease, kidney disease, liver disease, asthma, gastrointestinal disease, cancers, stroke, depression, back pain or knee pain, rheumatoid arthritis, others ( )	Check
3. Select your activities of daily living (modified Rankin Scale; mRS) 3-1. (mRS 0–1) No symptoms or No significant disability. Able to carry out all usual activities, despite some symptoms. 3-2. (mRS 2) Slight disability. Able to look after own affairs without assistance, but unable to carry out all activities. 3-3. (mRS 3) Moderate disability. Requires some help, but able to walk unassisted. 3-4. (mRS 4) Moderate severe disability. Unable to attend to own bodily needs without assistance, and unable to walk unassisted.	Check
4. In these three months, how many days per month does your headache occur?	( ) days/month
5. Does your headache have the following characteristics?	
5-1. unilateral location	Yes or No
5-2. pulsating quality	Yes or No
5-3. moderate or severe pain intensity	Yes or No
5-4. aggravation by or causing avoidance of routine physical activity	Yes or No
5-5. nausea and/or vomiting	Yes or No
5-6. photophobia and phonophobia	Yes or No
5. How long does your headache last?	( ) h or days
6-1. What do you use for headaches as acute medication?	(free answer)
6-2. How many days per month do you use such acute medication?	( ) days/month
7-1. Do you use prophylactic medication for headaches?	Yes or No
7-2. What prophylactic medication do you use?	(free answer)

Valid responses were those that filled in all the items in the questionnaire sheet. People who could not understand the questionnaire due to dementia, psychiatric disorder, mental retardation, and who indicated that they did not want to participate in this study, were excluded. The questionnaire sheets with one or more blank answers were also excluded from this study.

**Table 2 jcm-11-04707-t002:** Prevalence of headache disorders.

		Prevalence of Chronic Headache			Prevalence of Migraine			Prevalence of MOH		
	Total respondents’ number	Number; % (95%CI)			Number; % (95%CI)			Number; % (95%CI)		
Total	2858	342	11.97% (10.83–13.20)		26	0.91% (0.62–1.34)		20	0.70% (0.45–1.09)	
mRS 0–1	2384	310	13.00% (11.71–14.41)		24	1.01% (0.67–1.50)		19	0.80% (0.50–1.25)	
mRS 2	236	22	9.32% (6.18–13.77)		0	0% (0–1.92)		0	0% (0–1.92)	
mRS 3	218	8	3.67% (1.75–7.19)		2	0.92% (0.03–3.50)		1	0.45% (0–2.82)	
mRS 4	20	2	10.00% (1.57–31.32)		0	0% (0–18.98%)		0	0% (0–18.98%)	
All individuals with mRS 0–4										
Sex Women: Men (%Women)	1443:1415 (50.49%)	274:68 (80.11%)			24:2 (92.30%)			18:2 (90.00%)		
Age (median, IQR) (y.o.)	72 (69–77)	70 (67–74)		%Women	70 (67–75)		%Women	68 (65–70)		%Women
65–69	906	160	17.66% (15.31–20.28)	78.75%	12	1.32% (0.73–2.33)	100.00%	8	0.88% (0.41–1.77)	87.50%
70–74	956	98	10.25% (8.48–12.34)	83.67%	6	0.62% (0.25–1.40)	66.67%	11	1.15% (0.62–2.08)	90.90%
75–79	484	34	7.02% (5.05–9.68)	94.11%	6	1.23% (0.50–2.75)	100.00%	0	0% (0–0.95)	-
80–84	298	36	12.08% (8.82–16.03)	61.11%	0	0% (0–1.53)	-	1	0.34% (0–2.07)	100%
85–89	148	4	2.72% (0.82–6.98)	50.00%	0	0% (0–3.04)	-	0	0% (0–3.04)	-
90≤	66	10	15.15% (8.24–25.89)	100.00%	2	3.03% (0.22–11.01)	100.00%	0	0% (0–6.58)	-
Independent individuals with mRS 0–2										
Sex Women: Men (%Women)	1297:1323 (49.50%)	266:66 (80.12%)			22:2 (91.67%)			17:2 (89.47%)		
Age (median, IQR) (y.o.)	72 (68–76)	70 (66–76)		%Women	70 (67–72)		%Women	70 (68–73)		%Women
65–69	898	158	17.59% (15.23–20.23)	78.48%	12	1.34% (0.74–2.35)	100.00%	8	0.89% (0.42–1.78)	87.50%
70–74	916	98	10.70% (8.85–12.87)	83.67%	6	0.66% (0.26–1.46)	66.67%	11	1.20% (0.64–2.17)	90.90%
75–79	454	34	7.49% (5.38–10.30)	94.12%	6	1.32% (0.54–2.92)	100.00%	0	0% (0–1.01)	-
80–84	228	32	14.04% (10.08–19.18)	62.50%	0	0% (0–2.00)	-	0	0% (0–2.00)	-
85–89	90	2	2.22% (0.13–8.23)	0%	0	0% (0–4.91)	-	0	0% (0–4.91)	-
90≤	34	8	23.52% (12.20–40.23%)	100.00%	0	0% (0–12.07)	-	0	0% (0–12.07)	-

Abbreviations: CI; confident interval, IQR; interquartile range, MOH; medication-overuse headache.

**Table 3 jcm-11-04707-t003:** Characteristics of migraine, MOH, and other headache disorders among the independent elderly with mRS 0–2.

	Migraine (*n* = 24)	% (95%CI)	%Women	MOH (*n* = 19)	% (95%CI)	%Women	Other Headache Disorders (*n* = 292)	% (95%CI)	%Women
Age (median, IQR) (y.o.)	70 (67–72)			68 (65–70)			70 (68–74)		
Sex Women: Men (%Women)	22:2		91.67%	17:2		89.47%	229:63		78.42%
Headache occurring on (days/month)	8 (2–16)			28 (16–30)			1 (0.25–3)		
Low-frequency episodic migraine (0–3 days/month)	12	50% (31.42–68.57)	91.67%	0	-	-	242	82.59% (77.81–86.52)	76.45%
High-frequency episodic migraine (4–7 days/month)	0	0% (0–16.31)	100.00%	0	-	-	16	5.46% (3.33–8.75)	87.50%
Very high-frequency episodic migraine (8–14 days/month)	4	16.67% (6.07–36.47)	100.00%	0	-	-	14	4.78% (2.80–7.93)	85.71%
Chronic headache (15–29 days/month)	7	29.17% (14.71–49.38)	100.00%	10	52.63% (31.70–72.67)	90.00%	10	3.41% (1.78–6.25)	100.00%
Everyday	1	4.16% (0–21.87)	0%	9	47.37% (27.33–68.30)	88.89%	10	3.41% (1.78–6.25)	80.00%
Headache disorders characteristics									
Unilateral location	24	100.00% (83.69–100.00)	91.67%	9	47.37% (27.33–68.30)	88.89%	100	34.12% (28.93–39.74)	82.00%
Pulsating quality	16	66.67% (46.58–82.16)	87.50%	5	26.32% (11.45–49.15)	80.00%	47	16.04% (12.26–20.70)	82.98%
Moderate or severe pain intensity	8	33.33% (17.84–53.42)	100.00%	5	26.32% (11.45–49.15)	100.00%	21	7.17% (4.68–10.76)	90.48%
Aggravation by or causing avoidance of routine physical activity	8	33.33% (17.84–53.42)	100%	1	5.26% (0–26.48)	100.00%	20	6.82% (4.40–10.36)	100.00%
Nausea and/or vomiting	14	58.33% (38.81–75.56)	85.71%	2	10.53% (1.70–32.63)	50.00%	4	1.36% (0.40–3.56)	100.00%
Photophobia and phonophobia	12	50.00% (31.43–68.57)	100%	4	21.06% (7.95–43.89)	100%	6	2.05% (0.84–4.50)	100.00%
Migraine	-	100%	80.77%	3	15.79% (4.68–38.40)	66.67%	0	0.00% (0–1.56)	-
Duration (h)	24 (4–24)			2 (1–24)		2 (1–8)			
Less than 60 min	0	0% (0–16.31)	-	0	0% (0–19.79)	-	4	1.36% (0.40–3.56)	50.00%
1–3 h	0	0% (0–16.31)	-	10	52.63% (31.70–72.67)	100.00%	186	63.48% (57.82–68.79)	79.57%
4–8 h	6	25.00% (11.69–45.20)	100.00%	0	0% (0–19.79)	-	30	10.23% (7.23–14.28)	86.67%
9–24 h	18	75% (54.79–88.31)	88.89%	9	47.37% (27.33–68.30)	77.78%	56	19.16% (15.00–24.02)	66.07%
2–3 days	0	0% (0–16.31)	-	0	0% (0–19.79)	-	16	5.46% (3.33–8.75)	100.00%
Frequency of acute medication oral intake (days/months)	1 (0–2)			20 (10–30)			0 (0–2)		
<1 days/month	11	45.83% (27.88–64.93)	90.91%	0	-	-	155	52.90% (47.18–58.54)	77.42%
1–2 days/month	8	33.33% (17.84–53.42)	100.00%	0	-	-	74	25.26% (20.61–30.54)	72.62%
3–4 days/month	2	8,33% (1.16–27.00)	100.00%	0	-	-	35	11.94% (8.68–16.19)	88.57%
5–9 days/month	0	0% (0–16.31)	-	0	-	-	14	4.78% (2.80–7.93)	85.71%
10–14 days/month	1	4.17% (0–21.87)	100.00%	6	31.58% (15.16–54.20)	83.33%	5	1.76% (0.61–4.05)	80.00%
15–29 days/month	1	4.17% (0–21.87)	100.00%	4	21.06% (7.95–43.89)	100.00%	8	2.73% (1.30–5.39)	100.00%
Everyday	1	4.17% (0–21.87)	0%	8	42.11% (23.11–63.76)	88.89%	1	0.34% (0–2.11)	100.00%
Use of prophylactic medication	1	4.17% (0–21.87)	100.00%	0	0% (0–19.79)	-	1	0.34% (0–2.11)	100.00%

Abbreviations: CI; confident interval, IQR; interquartile range, MOH; medication-overuse headache

**Table 4 jcm-11-04707-t004:** Means of each variable among clusters.

Non-Hierarchical Clustering Using k-Means++							
	Total	Cluster 1 (Blue)	Cluster 2 (Green)	Cluster 3 (Orange)	Cluster 4 (Red)	Kruskal-Wallis Test or Chi-Squared Test	Mann-Whitney U Test or Chi-Squared Test
Number	332	239	27	16	50		
Age (y.o.)	71.66	71.97	70.33	70.25	71.34	0.559	-
Sex: %Women	0.80	0.81	0.89	1.00 ▲	0.66 ▽	0.009 **	Cramer’s V = 0.187
Diabetes	0.09	0.10	0.07	0	0.08	0.565	-
Hypertension	0.35	0.33	0.37	0.38	0.44	0.482	-
Dyslipidemia	0.19	0.20	0.15	0.50 ▲	0.06 ▽	0.001 **	Cramer’s V = 0.220
Heart disease	0.07	0.07	0.04	0.13	0.06	0.730	-
Kidney disease	0.01	0.02	0	0	0	0.665	-
Liver disease	0.01	0.02	0	0	0	0.665	-
Asthma	0.05	0.06	0	0.13	0	0.089	-
Gastrointestinal disease	0.10	0.10	0.11	0	0.08	0.549	-
Cancers	0.06	0.07	0	0.13	0	0.325	-
Stroke	0.04	0.02 ▽	0.07	0.25 ▲	0.04	<0.001 ***	Cramer’s V = 0.273
Depression	0.01	0.01	0	0.13 ▲	0	<0.001 ***	Cramer’s V = 0.235
Back pain, knee pain	0.14	0.14	0.22	0	0.18	0.203	-
Rheumatoid arthritis	0.01	0.01	0	0	0.04	0.253	-
Headache frequency (days/month)	4.69	2.27	27.56	2.14	4.71	<0.001 ***	*p* < 0.001 *** for 0,2,3 < 1; *p* < 0.001 *** for 0 < 3.
Unilateral location	0.39	0.34 ▽	0.41	0.38	0.62 ▲	0.004 **	Cramer’s V = 0.201
Pulsating quality	0.20	0.16 ▽	0.33	0.38	0.26	0.021 *	Cramer’s V = 0.171
Nausea and/or vomiting	0.05	0.04	0.07	0	0.12	0.109	-
Photophobia and phonophobia	0.06	0.03 ▽	0.15 ▲	0	0.05 ▲	<0.001 ***	Cramer’s V = 0.223
Aggravation by or causing avoidance of routine physical activity	0.08	0.05 ▲	0.10	0.38 ▲	0.14	<0.001 ***	Cramer’s V = 0.265
Moderate or severe pain intensity	0.10	0.07 ▽	0.22 ▲	0.38 ▲	0.12	<0.001 ***	Cramer’s V = 0.249
Duration (h)	9.02	2.47	16.11	48.00	24.00	<0.001 ***	*p* < 0.001 *** for 0 < 1.2.3; *p* = 0.002 ** for 1 < 2; *p* = 0.022 * for 1 < 3.
Frequency of acute medication oral intake (days/months)	2.92	2.00	14.56	3.44	0.88	<0.001 ***	*p* < 0.001 *** for 0.3 < 1.
Use of prophylactic medication is 1	0.01	0	0	0	0.02	0.567	-

*; *p* < 0.050, **; *p* < 0.010 ***; *p* < 0.001, ▲; significantly greater than expected frequency, ▽; significantly smaller than expected frequency.

**Table 5 jcm-11-04707-t005:** Previous reports on headache disorders prevalence in elderly.

Country	Year	Author	Methods	Sample Size of the Elderly	Age (y.o.)	Term	Headache Disorders Prevalence	CDH Prevalence	CM Prevalence	MOH Prevalence	EM Prevalence	TTH Prevalence
Population-based survey												
Japan	1997	Sakai [7]	Telephone survey	866	60≤	1 year	25%	-	-	-	About 1–9%	-
Taiwan	2000	Wang [17]	Face-to-face interview	1533	65≤	1 year	-	3.90%	0.98%	0.98%	-	-
Italy	2001	Prencipe [14]	Face-to-face interview then clinical evaluation by neurologists	833	65≤	1 year	51%	4.40%	-	1.70%	11.00%	44.50%
Japan	2004	Takeshima [6]	Face-to-face interview	2321	60≤	1 year	21.80%	-	-	-	2.15%	17.92%
Georgia	2009	Katsarava [16]	Face-to-face interview	26–140	65≤	1 year	-	-	-	-	0–5%	10–20%
Brazil	2010	Silva [20]	Face-to-face interview then clinical evaluation by neurologists	282	60<	1 year	-	3.90%	-	-	-	-
Sweden	2011	Jonsson [12]	Telephone survey	12,470	65≤	2 years	-	-	-	1.11%	-	-
Iran	2013	Shahbeigi [15]	Face-to-face interview	193	65≤	1 year	76%	53.40%	-	2.60%	6.70%	40.40%
Denmark	2014	Westergaard [21]	National health survey using questionnaire	16,344	65≤	3 months	-	2.98%	-	1.52%	-	-
Korea	2014	Park [22]	Face-to-face interview	184	60–69	1 year	-	3.80%	-	-	-	-
Denmark	2020	Westergaard [19]	National health survey using questionnaire	14,752	65≤	3 months	-	1.69%	-	1.28%	-	-
Japan	2022	Ours	Questionnaire during COVID-19 vaccination	2858	65≤	3 months	11.97%	1.57%	-	0.70%	0.91%	-
Headache clinic survey												
France	2018	Rijk [32]	Migraineurs in the headache clinic dataset	10	65≤	-	100%	44.44%	11%	33%	56%	-
Iran	2022	Togha [33]	Headache clinic dataset	172	60<	-	100%	-	19.20%	19.20%	16.90%	27.30%
Insurance data survey												
Dutch	2011	Dekker [34]	National insurance board database	1,422,554	60<	-	-	-	-	1181 (14.30%) triptan overuse of 8256 triptan users	8256 (0.580%) triptan users of 1422554 persons	-
Austria	2018	Zebenholzer [35]	Nationwide healthcare claims data	1,285,044	66–99	0.25 year	-	-	-	164 (9.21%) triptan overuse of 1779 triptan users	1779 (0.138%) triptan users of 1285044 persons	-

Abbreviations: CDH; chronic daily headache, CM; chronic migraine, EM; episodic migraine, MOH; medication-overuse headache, TTH; tension-type headache, y.o.; years old.

## Data Availability

Not applicable.

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
