# Peer review of "Questionnaire-Based Survey during COVID-19 Vaccination on the Prevalence of Elderly’s Migraine, Chronic Daily Headache, and Medication-Overuse Headache in One Japanese City—Itoigawa Hisui Study"

_jcm, 2022, doi:10.3390/jcm11164707_

Round 1

Reviewer 1 Report

The authors conducted a study of an interesting topic which is “  Questionnaire-based survey during COVID-19 vaccination on  the prevalence of elderly’s migraine, chronic daily headache, and medication-overuse headache in Japanese one city -Itoigawa Hisui Study“.  The quality of presentation of the manuscript, the interest of the readers and the scientific soundness are all high scored. Methodology and results are clearly presented. The manuscript is prepared based on almost the full capacity of the journal's requirements guidelines. 

         line 23-24: Correspondence person's data should include   email address and Tel/ fax number only. Please remove  first and last name, full postal address.

         line 105-107 , (quote) “We handed the questionnaire sheet and a pen to the citizens who underwent third- time vaccination, and they read it and wrote down the answers. Valid responses were those that filled in all the items on the questionnaire sheet “. It seems the above quoted should be re-phrased having in mind differences between phrase “wrote down the answer” and “filled in all the items on the questionnaire sheet “. It must be clearly explained that participants wrote down the answers on their own will and/ or participants marked already prepared answers only; or both of them were combined?

         A paragraph should be established on the inclusion and exclusion criteria in order to better continue reading scientifically and focus on such issues.

         Acknowledgments ( quote) “ We are thankful for the medical staff and servant services to support our work and data acquisition” should be more specific in reference to mention institution affiliated with “the medical staff and servant services.”

It was an excellent idea of the authors to add form of visualization such as  the Supplementary Materials Video S1: The clusters are plotted in the three-dimensional space, of which the axes were calculated by the principal component analysis consisting of the  variables.

Author Response

Thank you for your kind reviews. We revised our manuscript according to the reviewer’s suggesstions.

Reviewer 1

The authors conducted a study of an interesting topic which is “ Questionnaire-based survey during COVID-19 vaccination on  the prevalence of elderly’s migraine, chronic daily headache, and medication-overuse headache in Japanese one city -Itoigawa Hisui Study“.  The quality of presentation of the manuscript, the interest of the readers and the scientific soundness are all high scored. Methodology and results are clearly presented. The manuscript is prepared based on almost the full capacity of the journal's requirements guidelines. 
→Thank you for your kind review and comments.

line 23-24: Correspondence person's data should include  email address and Tel/ fax number only. Please remove  first and last name, full postal address.

→We deleted first name, last name, and full postal address. (Line 23-24)

line 105-107 , (quote) “We handed the questionnaire sheet and a pen to the citizens who underwent third- time vaccination, and they read it and wrote down the answers. Valid responses were those that filled in all the items on the questionnaire sheet “. It seems the above quoted should be re-phrased having in mind differences between phrase “wrote down the answer” and “filled in all the items on the questionnaire sheet “. It must be clearly explained that participants wrote down the answers on their own will and/ or participants marked already prepared answers only; or both of them were combined?

→We revised as follows; We handed the questionnaire sheet and a pen to the citizens who underwent third-time vaccination, and they read it, and wrote down the answers or filled in all the items on the questionnaire sheet. The items we asked are shown in Table 1 in the next paragraph. (Line 115-116)

A paragraph should be established on the inclusion and exclusion criteria in order to better continue reading scientifically and focus on such issues.

→We added the paragraph regarding the inclusion and exclusion criteria; The inclusion criteria were; people aged over 64 with valid responses that filled in all the items on the questionnaire sheet. The exclusion criteria were; people who could not understand the questionnaire due to dementia, psychiatric disorder, mental retardation, people who indicated that they did not want to participate in this study, and people with invalid responses as the questionnaire sheets with one or more blank answers. People aged over 64 who met these criteria were analyzed. Among those, headache, migraine, CDH, MOH prevalence, and their relationship to the items in the questionnaire sheet were investigated.(Line 117-126)

Acknowledgments ( quote) “ We are thankful for the medical staff and servant services to support our work and data acquisition” should be more specific in reference to mention institution affiliated with “the medical staff and servant services.”
→We rewrote as follows; We are thankful for the medical staff in the Itoigawa General Hospital, Nou National Insurance Hospital and servant services in Itoigawa City to support our work and data acquisition. (Line 516-518)

 It was an excellent idea of the authors to add form of visualization such as  the Supplementary Materials Video S1: The clusters are plotted in the three-dimensional space, of which the axes were calculated by the principal component analysis consisting of the  variables.
→Thank you for your kind comments.

Reviewer 2 Report

Whereas this is a nice idea about finding more about the headache disorders in the elderly. The study has many design flaws and the both of the results and the manuscript need extensive revisions that will not be applicable if the study will not be conducted correctly.

1.       Do not use any abbreviations in the abstract

2.       Page 2 Line 46: The 2 representative headaches: This in not the case. It is only if we refer to primary headache disorders. Please change accordingly.

3.       Page 2 line 49: migraine headaches: change to migraine attacks

4.       Page 2 Lines 57-58: organic diseases: Migraine is an “organic” disease and not psychogenic. Should you refer to secondary causes?

5.       Page 2 Line 58 detailed primary headache: What do you mean by that?

6.       Page 2 Lines 65-68: Insufficient use of English. Please rephrase.

7.       Page 2 Line 75: The headache: Delete “the”. Furthermore, what do you mean by chronic daily headache? New persistent daily headache, chronic migraine or chronic tension type headache or MOH? Is it all together? Please explain.

8.       Page 2 Line 80: diagnose all headache: change to diagnose all headache disorders.

9.       Page 3 Line 108: dementia, psychiatric disorder, mental retardation: Did you screen for these disorders? How did you exclude them? Did you do a mental examination testing or by medical history?

10.   Table 1. What was the rationale behind question 3? Why did you use those specific medical problems? Why not for example COPD or sleep apnea that are correlated with headache disorders?

11.   Page 4 Line 133: chronic headache: That is not in the ICHD-3 criteria, only that there are at least 5 attacks.

12.   Page 4 Lines 138-140: MOH is Headache occurring on 15 or more days/month in a patient with a pre-existing primary headache and developing as a consequence of regular overuse of acute or symptomatic headache medication (on 10 or more or 15 or more days/month, depending on the medication) for more than 3 months. You do not mention the duration which important.

13.   Page 4 Lines 142-144: CDH includes many chronic headache disorders. Please analyse that.

14.   Change all headache and headache to all headache disorders and headache disorders thought the text.

15.   Page 5 Line 186: all mRS: change to all mRS scores.

16.   Page 17 Limitations: You have not mentioned the secondary headaches which must be excluded by definition to reach a diagnosis of a primary headache disorder. Which medications that the elderly have already taken that can cause headache resembling either TTH or migraine. What about the vaccination itself? Or even the post-COVID-19 chronic headache? There have not been mentioned.

Author Response

Thank you for your kind reviews. We revised our manuscript according to the reviewer’s suggestions.

Whereas this is a nice idea about finding more about the headache disorders in the elderly. The study has many design flaws and the both of the results and the manuscript need extensive revisions that will not be applicable if the study will not be conducted correctly.
→Thank you for your constructive comments and we are grateful for your kind review. We revised our manuscript according to your suggestions.

  1. Do not use any abbreviations in the abstract
    →We rewrote the abstracts without abbreviations of COVID-19 and ICHD-3. (Abstract)

  2. Page 2 Line 46: The 2 representative headaches: This in not the case. It is only if we refer to primary headache disorders. Please change accordingly.
    →We rewrote as “primary headache.” (Line 48)

  3. Page 2 line 49: migraine headaches: change to migraine attacks
    →We rewrote as “migraine attacks.” (Line 51)

  4. Page 2 Lines 57-58: organic diseases: Migraine is an “organic” disease and not psychogenic. Should you refer to secondary causes?
    →We rewrote as “secondary headaches” (Line 60)
  5. Page 2 Line 58 detailed primary headache: What do you mean by that?
    →We rewrote as “the appropriate diagnosis of primary headache subtypes and their treatment is insufficient” (Line 60-61)
  6. Page 2 Lines 65-68: Insufficient use of English. Please rephrase.
    →We rephrased as; Also, other diseases causing headaches, such as transient ischemic attacks and amyloid angiopathy, and the presence of multiple comorbidities and polypharmacy make the elderly’s headache disorders complicated (Line 71-73)
  7. Page 2 Line 75: The headache: Delete “the”.
    →We deleted it. (Line 80)

Furthermore, what do you mean by chronic daily headache? New persistent daily headache, chronic migraine or chronic tension type headache or MOH? Is it all together? Please explain.
→We described as: CDH in this context means as a case with headaches occurring at least 15 days per month for 3 or more consecutive months [14,15,17–22]. (Line 82-84)

  1. Page 2 Line 80: diagnose all headache: change to diagnose all headache disorders.
    →We rewrote as you suggested.
  2. Page 3 Line 108: dementia, psychiatric disorder, mental retardation: Did you screen for these disorders? How did you exclude them? Did you do a mental examination testing or by medical history?
    →Thank you for your pointing out. We did not perform these tests. We described as follows; The diagnosis of dementia, psychiatric disorders, and mental retardation was self-reported by respondents or their families. (Line 122-124)
  3. Table 1. What was the rationale behind question 3? Why did you use those specific medical problems? Why not for example COPD or sleep apnea that are correlated with headache disorders?
    →Thank you for your constructive suggestions. We did not think of using these diseases as the questionnaire items, but we found them in the free answer. However, the number of these diseases was very small, so we did not describe them. We discuss this in the limitation section; Seventh, we did not ask about the presence of chronic obstructive pulmonary disease and sleep apnea, which is known to be related to headaches. (Line 487-489)
  4. Page 4 Line 133: chronic headache: That is not in the ICHD-3 criteria, only that there are at least 5 attacks.
    →Thank you for your pointing out. We rewrote as follows; In the valid respondents, cases with headaches within the last 3 months were considered headaches sufferers. Migraine and MOH were defined in compliance with the ICHD-3 [3] as much as possible in the questionnaire sheet, but not strictly the same as described in the ICHD-3. (Line 147-149)
  5. Page 4 Lines 138-140: MOH is Headache occurring on 15 or more days/month in a patient with a pre-existing primary headache and developing as a consequence of regular overuse of acute or symptomatic headache medication (on 10 or more or 15 or more days/month, depending on the medication) for more than 3 months. You do not mention the duration which important.
    →We write limitation about it as follows; Fifth, the headache diagnosis was not strictly based on the ICHD-3; We did not investigate whether there were at least 5 migraine attacks or not. Also, MOH is a headache occurring on 15 or more days/month in a patient with a pre-existing primary headache and developing as a consequence of regular overuse of acute or symptomatic headache medication (on 10 or more or 15 or more days/month, depending on the medication) for more than 3 months. However, we did not investigate the duration of regular overuse. (Line 477-483)
  6. Page 4 Lines 142-144: CDH includes many chronic headache disorders. Please analyse that.
    →We described as; Of the 45 CDH respondents, nobody had migraine, but 9 (20.00%) had MOH.” in all mRS score respondents (Line 208-209) and “Of the 45 CDH respondents, nobody had migraine, but 9 (20.45%) had MOH.” in mRS score 0-2 respondents. However, we could not diagnose the ICHD-3 code based on the questionnaire sheets (Line 213-214).
  7. Change all headache and headache to all headache disorders and headache disorders thought the text.
    →We changed as you suggested.
  8. Page 5 Line 186: all mRS: change to all mRS scores.
    →We changed as you suggested. (Line 203)
  9. Page 17 Limitations: You have not mentioned the secondary headaches which must be excluded by definition to reach a diagnosis of a primary headache disorder. Which medications that the elderly have already taken that can cause headache resembling either TTH or migraine. What about the vaccination itself? Or even the post-COVID-19 chronic headache? There have not been mentioned.
    →We described in the limitation as follows; Sixth, we could not investigate the possibility of other secondary headaches by the questionnaire sheet, which must be excluded by definition to reach a diagnosis of primary headaches. Also, the use of medicines with headache side effects was not investigated. Furthermore, COVID-19 vaccination-related headaches and post-COVID-19 headaches were not investigated. (Line 483-487)

Reviewer 3 Report

Thank you for submitting the manuscript. I have read your paper with great attention and interest. The topic is certainly of interest to readers. Your research is interesting and well structured. However, I ask you for some revisions.

First of all, the title should be restructured.In fact, at a first little careful reading it might seem that your paper deals with a correlation between headache and covid, or between headache and vaccine. Instead, you only used vaccination as an opportunity to submit the survey.

I ask you to check the key words, making them appropriate to the Mesh terms

You have carried out your survey among the elderly who have undergone vaccination. I think it is necessary to clarify the Japanese context. In fact, it is necessary to clarify whether the vaccine in Japan was mandatory or recommended. This changes the nature of the champion enormously. In fact, the reading of the data of those who have decided not to answer changes on the basis of the voluntary presence or obliged to vaccinate. In fact, a different sensitivity towards one's own health emerges.

At one point you assert that headache treatments in Japan are inadequate. However, there are no references or explanations. I ask you to clarify better. Furthermore, you should relate this information to the limitation you emphasize regarding the rural reality in which you conducted the survey.The problem of patchy pain therapy across a nation and organizational and resource differences is a factor affecting the quality of patient care. I suggest you expand this theme in the discussion and on this subject I suggest references to make international comparisons:

doi: 10.1185/03007995.2013.861349.

doi: 10.2147/JPR.S328434.

  • DOI: 10.1016/j.berh.2004.04.004
  • I therefore ask you to rephrase the introduction, discussion and conclusions in the light of my suggestions. I hope these comments are helpful to you.

Kind Regards

  •  
  •  

Author Response

Thank you for your kind reviews. We revised our manuscript according to the reviewer’s suggestions.

Thank you for submitting the manuscript. I have read your paper with great attention and interest. The topic is certainly of interest to readers. Your research is interesting and well structured. However, I ask you for some revisions.
→Thank you for your kind review and suggestions.

First of all, the title should be restructured. In fact, at a first little careful reading it might seem that your paper deals with a correlation between headache and covid, or between headache and vaccine. Instead, you only used vaccination as an opportunity to submit the survey.
→We changed the title as “Questionnaire-based survey on the prevalence of elderly’s migraine, chronic daily headache, and medication-overuse headache in Japanese one city -Itoigawa Hisui Study-.“

I ask you to check the key words, making them appropriate to the Mesh terms
→We changed as: aged; artificial intelligence; chronic daily headache (CDH); cluster analysis; epi-demiology; medication-overuse headache (MOH); migraine; prevalence

You have carried out your survey among the elderly who have undergone vaccination. I think it is necessary to clarify the Japanese context. In fact, it is necessary to clarify whether the vaccine in Japan was mandatory or recommended. This changes the nature of the champion enormously. In fact, the reading of the data of those who have decided not to answer changes on the basis of the voluntary presence or obliged to vaccinate. In fact, a different sensitivity towards one's own health emerges.
→We described as follows; It is not mandatory, so people had the right to refuse. However, the third-time COVID-19 vaccination coverage among the elderly was 90.1% on July 20th (https://www.kantei.go.jp/jp/headline/kansensho/vaccine.html). (Line 105-107)

At one point you assert that headache treatments in Japan are inadequate. However, there are no references or explanations. I ask you to clarify better.
→We mention the inadequacy in the introduction and refer to some articles ([5] and [11]). (Line 341)

Furthermore, you should relate this information to the limitation you emphasize regarding the rural reality in which you conducted the survey.The problem of patchy pain therapy across a nation and organizational and resource differences is a factor affecting the quality of patient care. I suggest you expand this theme in the discussion and on this subject I suggest references to make international comparisons:
doi: 10.1185/03007995.2013.861349.
doi: 10.2147/JPR.S328434.
DOI: 10.1016/j.berh.2004.04.004
→Thank you for your constructive suggestions. We described in the discussion section as follows;
4.6 Insufficient medical resources for headache treatment
Itoigawa city, where this study was performed, is rural, and the medical resources are not rich. In rural areas with limited medical resources, the potential misuse of pain and headache clinics can be raised, seeing them as “last resorts” for hopeless cases [53]. Lack of appropriate treatment for chronic pain remains an ongoing major healthcare problem also in other countries. In Germany, Italy, and Turkey, doctors treating chronic cancer and musculoskeletal pain feel the need to change treatment for a third of the patients. They think of changing the pain medication (52.4%), co-medication (42.2%), and non-medical therapy (19.0%) [54]. Therefore, improved communication between patients and doctors is still needed to find the best medications [55]. From the Italian survey, doctors treating musculoskeletal pain do not frequently use multidimensional questionnaires to evaluate patients’ status, which assess physical, social activities, psychological distress, sleep, enjoyment of life, and functioning. This insufficient medical attitude towards pain treatment among primary care providers suggests the need of further efforts toward a pain management-oriented education for doctors and developed medical coordination systems between general practitioners and second-level pain centers [56]. These problems are also seen in Japan [57]. Also, a survey in Japan showed that over half of Japanese people incorrectly believe that they should be endure pain, and that over 80% of them are unaware of appropriate ways to cope with pain [58]. About 20% of headache sufferers feel that doctors would not be kind to them if they consulted them. About 41% of them do not know the presence of prophylactic medications. To solve these problems, a headache awareness campaign for patients, non-headache sufferers, and doctors has also been performed [59]. However, the insufficient medical resources in rural Itoigawa and doctors’ and patients’ inappropriate at-attitudes toward their headaches may have influenced our survey result; this is one of the limitations. (Line 440-464)

I therefore ask you to rephrase the introduction, discussion and conclusions in the light of my suggestions. I hope these comments are helpful to you. Kind Regards
→Thank you for your constructive suggestion. We added the paragraph described above in the discussion section. We did not change the conclusion because we believe that conclusion should be mention the fact which this study showed. We apologize for not meeting your expectations.

Round 2

Reviewer 2 Report

Thank you for considering the changes and suggestions.

Limitations of a manuscript are an important paragraph and should be transparent and analytic.

Author Response

Thank you for your kind comments. We rewrote the limitation section as follows;

4.7. Limitation

First, this study was performed in a rural city in Japan, and the respondents covered about only 20% of the elderly population. Itoigawa City does not yet have adequate neurological and headache disorders care resources, and the prevalence may vary compared to urban areas with more medical resources in Japan. Second, we performed this questionnaire at the 2 large vaccination sites, so we could not evaluate the citizens who did not come to the vaccination site or who selected other vaccination sites. Third, the recall bias can be present, affecting the prevalence results. Fourth, migraine and MOH should be diagnosed after filling a prospective 3-month headache diary. Our study investigated retrospectively, so the questionnaire-based diagnosis should be interpreted carefully. Also, the headache diagnosis was not strictly based on the ICHD-3; We did not investigate whether there were at least 5 migraine attacks or not. Also, MOH is a headache occurring on 15 or more days/month in a patient with a pre-existing primary headache and developing as a consequence of regular overuse of acute or symptomatic headache medication (on 10 or more or 15 or more days/month, depending on the medication) for more than 3 months. However, we did not investigate the duration of regular overuse. Fifth, the 4 clusters of the clustering results in this study were mathematically decided, so the clinical meanings of each cluster should be carefully interpreted. Sixth, we could not investigate the possibility of other secondary headaches by the questionnaire sheet, which must be excluded by definition to reach a diagnosis of primary headaches based on the ICHD-3. Also, the use of medicines with side effects of headaches was not investigated. Furthermore, COVID-19 vaccination-related headaches and post-COVID-19 headaches were not investigated. Seventh, we did not ask about the presence of chronic obstructive pulmonary disease and sleep apnea, which is known comorbidities related to headaches. A further large study on the elderly’s headache disorders prevalence all over Japan is needed.

Reviewer 3 Report

reference 56 is badly quoted, mdpi style must be respected: please correct. for the rest I am satisfied.

Kind Regards

Author Response

Thank you for your kind review. We changed as follows. Sorry for that.

  1. Vittori, A.; Petrucci, E.; Cascella, M.; Innamorato, M.; Cuomo, A.; Giarratano, A.; Petrini, F.; Marinangeli, F. Pursuing the recovery of severe chronic musculoskeletal pain in Italy: Clinical and organizational perspectives from a SIAARTI survey. J. Pain Res. 2021, 14, 3401–3410, doi:10.2147/JPR.S328434.